# GQA-Q2Q: A Large-scale Dataset for Resolving Entity Ambiguity in Visual Question-Answering via Clarifying Sub-Question

## Abstract

Vision-Language Models (VLMs) have achieved remarkable results on various visual question-answering (VQA) benchmarks. However, their performance is significantly impacted by ambiguous questions in which the target entity in the image is not clearly identified. To address and evaluate this issue, it is essential to create a dedicated benchmark dataset that aligns ambiguous questions with a clarifying sub-question. However, constructing a large, high-quality dataset is costly, particularly when it relies on expert annotations. To efficiently construct such a dataset at scale, this paper presents a hybrid human-machine pipeline. This pipeline begins with generating a small initial set of sub-questions using attribute-based templates, which are then refined through human annotation. This initial annotated set serves as the foundation for training a sub-question generator and a validator, and the generator and the validator together allow automatic construction of a large-scale dataset. As a result, this paper presents a new large-scale dataset, GQA-Q2Q, designed to disambiguate unclear entities through clarifying sub-questions. Furthermore, a VQA framework is introduced which utilizes the clarifying sub-questions to resolve entity ambiguity before producing a final answer. The experimental results demonstrate that this approach enhances VQA performance, validating the effectiveness of the proposed dataset.

## 1 Introduction

Visual question answering (VQA) aims to provide an accurate answer to a natural language question based on a given image related to the question. With the advancement of transformer-based vision-language models (VLMs), there have been remarkable performance improvements by the VQA models that adopt a VLM as a question-answering model across a variety of benchmarks (Bai et al., 2023; Liu et al., 2023). Despite the improvements, when a question mentions an unclearly specified entity, the VQA models often fail to derive an accurate answer from the image alone. For example, in Figure 1 sampled from GQA dataset (Hudson & Manning, 2019), the phrase '*happy man*' in the question is originally intended to refer to the man on the right side of the image. However, since there is another happy man on the left side, there are two possible answers although only one of them is actually correct. Therefore, it is necessary to clarify such an ambiguous entity, and it is an effective and practical approach to introduce sub-questions for clarification.

In order to cope with the entity ambiguity in the image, this paper defines an ambiguous question as the one that lacks a clearly identifiable target entity for accurate reasoning within the context of VQA. Based on this definition, it presents GQA-Q2Q (*GQA-ambiguous Question to clarified Question*), a large-scale dataset of 135K sub-questions for the ambiguous questions of GQA dataset. Since GQA-Q2Q is constructed based on the GQA dataset, it allows direct utilization of the scene graph information for precise entity grounding. The sub-questions in this dataset are formulated as yes/no ones, and they are designed to clarify which entity instance the original question refers to, thereby resolving the entity ambiguity of the original question. By leveraging a sub-question and its sub-answer as well as the original ambiguous question, a VQA model can arrive at a correct answer even when the original question mentions an ambiguous entity. Therefore, the construction of a large-scale, high-quality dataset of clarifying sub-questions is essential for training and evaluating VQA models that can manage ambiguous entities.

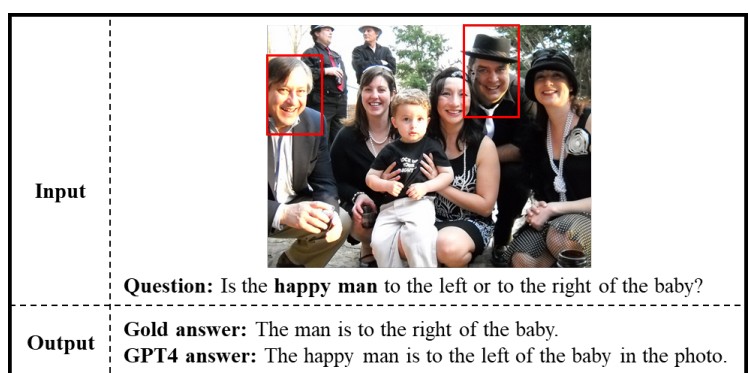

| Input | |
|---|---|
| | **Question:** Is the **happy man** to the left or to the right of the baby? |
| Output | **Gold answer:** The man is to the right of the baby.
**GPT4 answer:** The happy man is to the left of the baby in the photo. |

Figure 1: An example ambiguous question in GQA dataset.

Despite the necessity of such a dataset, the manual construction of a large number of sub-questions is extremely expensive. Thus, this paper proposes a human-machine collaborative pipeline to construct a large, high-quality dataset of entity-clarifying sub-questions. The pipeline begins with manual construction of a small initial dataset by a human annotator. For every ambiguous question, a simple template-based sub-question generator prepares plural candidate sub-questions, and a human annotator chooses the best and plausible sub-question among the candidates if there is any. The automatic large-volume construction of sub-questions is performed in a similar way to the human construction. In this automatic construction, a VLM-based sub-question generator replaces the template-based sub-question generator, since the performance of the template-based generator is not reliable enough. In addition, the human annotator is replaced with a sub-question validator. These sub-question generator and validator are first trained with the initial dataset. The sub-question generator generates plural candidate sub-questions by nucleus sampling (Holtzman et al., 2020), and the validator selects only the highly confident candidates among the generated candidate sub-questions as human annotators do. Due to the large volume of ambiguous questions of GQA dataset and the high performance of the sub-question generator and validator, this human-machine collaboration achieves the scale and the quality of the proposed dataset simultaneously.

To validate the effectiveness of GQA-Q2Q, a VQA framework is introduced, which consists of an ambiguity detector, a sub-question generator, a sub-question respondent, and a final answerer. The ambiguity detector is a classifier that determines whether an input question contains an ambiguous entity. If the question is not ambiguous, the final answerer immediately generates a final answer with the input image and the question. Otherwise, the sub-question generator generates sub-questions to clarify the ambiguous entity, where it is a fine-tuned VLM with GQA-Q2Q. Then, the sub-question respondent (an oracle or a model that is aware of the ground-truth target entity) provides sub-answers to the sub-questions. Finally, a final answerer generates a final answer by leveraging the input image, the original question, the sub-questions, and the sub-answers.

The main contributions of this paper are as follows:

- This paper focuses on the entity ambiguity in VQA questions and constructs a high-quality dataset of 135,846 sub-questions from GQA dataset. To the best of our knowledge, this is the first large-scale dataset designed to resolve the entity ambiguity of VQA through targeted sub-question generation.

- This paper develops an efficient pipeline for dataset collection. The sub-question generator and the validator in the pipeline are trained with the initial dataset created by attribute-based templates and human verification, and then they are used to construct high-confidence sub-questions automatically. This enables a large and high-quality sub-question collection.

- This paper proposes a novel VQA framework that explicitly incorporates ambiguity resolution by answering a clarifying sub-question before answering the main question, which leads to improved performance of VQA.

## 2 RELATED WORK

### 2.1 VISUAL QUESTION-ANSWERING

Visual question-answering (VQA) is a representative multi-modal task in which a model answers natural language questions about a given image. Many recent studies have introduced various VQA datasets encompassing diverse types of visual information (Ren et al., 2015; Antol et al., 2015; Goyal et al., 2017; Krishna et al., 2017; Marino et al., 2019; Wang et al., 2022). For instance, GQA dataset provides scenario-based question-answer pairs derived from real images, where the questions are generated from scene graphs and linguistic templates (Hudson & Manning, 2019). VQA models also have evolved alongside the development of benchmark datasets. Recently, the transformer-based models such as BLIP-2 (Li et al., 2023) and LLaVA (Liu et al., 2023) pretrained with large-scale data have achieved strong zero-shot performances in various VQA benchmarks (Goyal et al., 2017; Hudson & Manning, 2019; Marino et al., 2019).

### 2.2 QUESTION CLARIFICATION

Several studies have proposed question-clarification methods that resolve a question ambiguity in VQA. These can be broadly divided into two categories. One category is to generate a clarified version of an ambiguous question by removing ambiguity before passing it to a VQA model. Prasad et al. (2024) proposed a method for clarifying ambiguous questions by rephrasing questions, augmenting them with visual groundings, and reasoning an answer with a confidence score. The other category is to obtain additional information from a user via sub-questions, where the aim of sub-questions is to clarify the intent of an original question (Selvaraju et al., 2020; Wang et al., 2022). For example, Khan et al. (2023) improved zero-shot VQA performance by generating sub-questions through question decomposition. However, most previous studies do not define clearly what constitutes the ambiguity. Jian et al. (2025) proposed a benchmark called ClearVQA for clarifying ambiguities in visual questions. While ClearVQA includes 9,243 ambiguous questions primarily addressing text-based referential ambiguities, it remains relatively small in scale and does not explicitly focus on entity-level ambiguities grounded in visual content.

### 2.3 AUTOMATIC DATA CONSTRUCTION

Automatic data construction is an effective alternative to costly manual annotation for large-scale dataset creation for visual-language tasks. Some previous studies proposed large-scale datasets constructed with hand-crafted templates (Ren et al., 2015; Hudson & Manning, 2019). However, template-based data construction suffers from limited diversity and applicability, as it often produces patterned data that may not fully reflect real-world scenarios. With the recent advances in generative capability of LLMs that consider user instructions, data generation can be automated by using an LLM (Liu et al., 2022; Wiegreffe et al., 2022). Lee et al. (2023) presented a large-scale dataset of sensitive questions and acceptable responses, where the candidate questions and responses are generated by a machine and subsequently verified by human annotators. In this work, an initial dataset is collected through human annotations, and a data generation model and an evaluation model are trained with the annotated data to subsequently enable an automatic construction of a large-scale dataset.

## 3 PRELIMINARIES

An ambiguous question in text-based question answering is defined as the one that has multiple possible answers or no definite answer at all (Min et al., 2020). While this definition is valid also for visual question answering, visual ambiguous questions require additional multi-modal comprehension of images and texts. Prasad et al. (2024) proposed a method to handle ambiguous questions by reformulating the questions into more specific and detailed ones based on visually grounded information. However, the method has a limitation in that it just reformulates the questions, but does not resolve the ambiguity within the questions explicitly. Chen et al. (2025) presented VQ-FocusAmbiguity, a VQA dataset that grounds each question to its corresponding image region according to the concept of focus ambiguity (Chen et al., 2025). However, VQ-FocusAmbiguity does not provide a concrete approach for resolving the ambiguity.

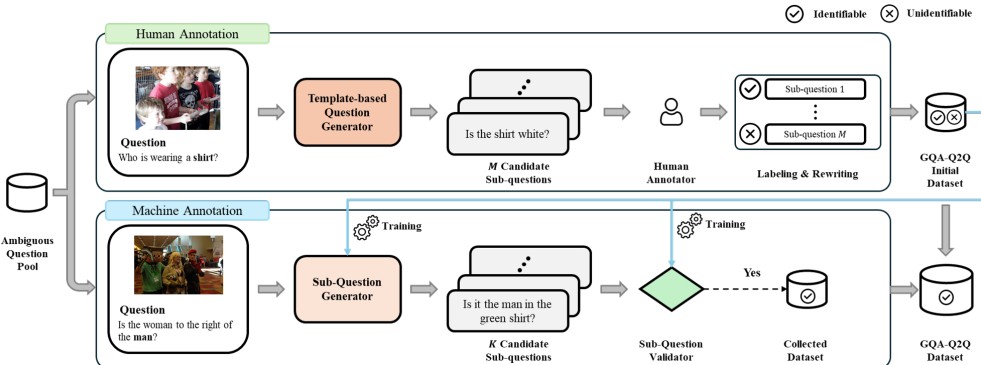

Figure 2: Overview of the GQA-Q2Q dataset construction process.

To effectively address ambiguous questions, it is essential to establish a clear criterion for determining whether a given question is ambiguous. In this paper, a question is defined as ambiguous if its entity corresponds to multiple instances within a given image. That is, a question is ambiguous if it exhibits *entity ambiguity*. Entity ambiguity is different from focus ambiguity in that it specifically addresses the problem of under-specified or ambiguous entities in VQA questions, while focus ambiguity refers more generally to situations that can be grounded to multiple regions of an image.

One possible solution for handling entity ambiguity is question clarification. This solution resolves ambiguity by asking a sub-question to an oracle or a questioner who is aware of the target entity. To eliminate ambiguity in a question, it is natural for humans to ask additional sub-questions that help clarify unclear parts of the original question. Thus, a good sub-question is the one that can uniquely identify the target instance in the image by referring to the distinguishable property of the entity. In this paper, the sub-questions are restricted to yes/no ones. Thus, receiving a '*yes*' answer to a sub-question implies that the sub-question indeed refers to the target instance. For instance, a good sub-question to identify '*happy man*', the ambiguous entity in Figure 1, could be "*Is the man wearing a hat*?" since only one among the two happy men wears a hat. After the clarification, a VQA model can answer the ambiguous question accurately by leveraging the sub-question and its sub-answer.

While sub-questioning is effective for resolving entity ambiguity in VQA, there is no available dataset designed for the benchmark of clarifying sub-questions. To fulfill this gap, this paper introduces GQA-Q2Q, a benchmark dataset for high-quality clarifying sub-questions. While manual construction of a dataset ensures high quality of the dataset, it is extremely expensive to construct it manually on a large scale. Recent advancements in large neural models showed that a machine trained with a small expert-annotated dataset can replace human experts effectively (Lee et al., 2023). Therefore, this paper employs a two-stage process for building GQA-Q2Q. The process begins with human annotation to develop a small initial dataset, and then enlarges the dataset with machine annotation. This leads to a dataset that achieves both scalability and high quality.

## 4 GQA-Q2Q DATASET

### 4.1 GQA DATASET AS A BASE DATASET

The proposed GQA-Q2Q dataset is based on GQA dataset, a widely used benchmark for visual question-answering. GQA dataset provides 113,018 images along with 22 million questions generated from scene graphs. Each scene graph contains information about an image including objects, attributes, and their relations. Moreover, since the target is annotated in each question, it is appropriate to identify the ambiguity of the entity of questions. This paper adopts the balanced version of GQA dataset, which contains 943,000 training, 132,062 validation, and 95,336 test questions, where the balanced version is obtained by downsampling unbalanced original dataset.

Recall that the questions that reference at least one entity appearing more than once in an image are ambiguous ones. Thus, a set of ambiguous questions, denoted as $\mathcal{Q}_{\text{amb}}$, is compiled from the

| Dataset | Train | Validation | Test |
|---|---|---|---|
| # of sub-questions | 58,646 | 7,297 | 7,337 |
| – Identifiable | 24,779 | 3,081 | 3,062 |
| – Unidentifiable | 33,867 | 4,216 | 4,275 |

Table 1: A simple statics on the initial GQA-Q2Q dataset.

| Disambiguity | | Fluency | |
|---|---|---|---|
| Human-annotated Data | Machine-annotated Data | Human-annotated Data | Machine-annotated Data |
| 2.66±0.18 | 2.42±0.20 | 2.84±0.17 | 2.80±0.19 |

Table 2: Human evaluations of GQA-Q2Q in terms of disambiguity and fluency

training questions, and it has 125,854 (13.34%) ambiguous questions. The adjective modifiers in these ambiguous questions are removed to increase their ambiguity, where the modifiers are identified by a constituency parser 'ptb-3-revised_electra-large' from Stanza library[1]. For example, the ambiguous question in Figure 1, "*Is the happy man to the left or to the right of the baby*?", becomes more ambiguous if the modifier '*happy*' is removed.

## 4.2 DATA COLLECTION OF GQA-Q2Q

Figure 2 illustrates how GQA-Q2Q dataset is constructed. The dataset is built in two stages. In the first stage, a small initial set is prepared through manual annotation. This initial set is used to train the sub-question generator and the validator for the next machine annotation stage. The sub-question generator generates a large scale of sub-questions from $\mathcal{Q}_{amb}$, and the validity of the generated sub-questions is ensured by the validator.

### 4.2.1 HUMAN ANNOTATION

For each question $q \in \mathcal{Q}_{amb}$, $M$ candidate sub-questions are generated using five linguistic templates. The templates are designed to ask about a unique attribute of a target entity to distinguish the target instance from other instances. In Figure 2, the target entity is '*shirt*' and one of the attribute values of the shirt is '*white*'. Thus, from the template "*Is* <target entity> <attribute value>?", a candidate sub-question "*Is the shirt white*?" is generated. A more detailed explanation about the templates is provided in Appendix A.1.

The sub-questions that can identify a target instance are called *identifiable* sub-questions, while those that cannot identify a target instance are *unidentifiable* sub-questions. Note that the automatically-generated candidates contain both types of sub-questions. Thus, the candidates are validated by a human annotator. A human annotator reviews the candidates for $q$, and selects among them the most appropriate sub-question that allows the target instance to be identified. If a human annotator selects one appropriate sub-question, the remaining $M - 1$ candidates can be both identifiable and unidentifiable.

If none of the candidates are appropriate, the annotator creates a new identifiable sub-question manually. In this case, all generated candidates are definitely unidentifiable for $q$. These definitely unidentifiable sub-questions are also included in the initial set for the use of the next machine annotation. The number of sub-questions is summarized in Table 1. There are 73,280 sub-questions. Among them, 26,510 sub-questions are identifiable, and the remaining 46,770 sub-questions are unidentifiable.

### 4.2.2 MACHINE ANNOTATION

While human annotation ensures high data quality, it is costly and requires expert annotators with a thorough understanding of a task. To address these limitations and efficiently construct a large-scale GQA-Q2Q dataset beyond the initial set, an automatic data construction is employed. The automatic construction follows the process of the manual construction of the initial set, but the main problem with the manual construction is the reliability of the linguistic templates. Therefore, in the machine annotation stage, a sub-question generator and a sub-question validator are proposed instead of the templates and a human annotator to ensure the quality of automatically-generated sub-questions.

---

[1]https://stanfordnlp.github.io/stanza

Algorithm 1 describes the process of automatic construction of GQA-Q2Q. Since the initial set, $\mathcal{D}_{init}$, is available, it is used to train a sub-question generator $f_{\text{SG}}(\cdot)$ and a validator $f_{\text{SV}}(\cdot)$ by regarding identifiable sub-questions as positive samples and unidentifiable sub-questions as negative samples. The sub-question generator $f_{\text{SG}}(\cdot)$ is trained with only identifiable sub-questions, while the validator $f_{\text{SV}}(\cdot)$ is trained with both identifiable and unidentifiable sub-questions. Recent large VLMs show strong performance on diverse tasks that require both visual and textual reasoning. Thus, LLaVA Liu et al. (2023) is adopted as a backbone model for implementing both $f_{\text{SG}}(\cdot)$ and $f_{\text{SV}}(\cdot)$.

---

**Algorithm 1** Automatic Data Construction for GQA-Q2Q

**Require:** $\mathcal{D}_{\text{init}}$ (initial human-annotated dataset), $\mathcal{D}_{\text{id\_init}}$ (initial set of identifiable sub-questions), $\mathcal{Q}_{\text{amb}}$ (ambiguous question pool), and $\tau$ (confidence threshold)
1: Train sub-question generator $f_{\text{SG}}$ and validator $f_{\text{SV}}$ using $\mathcal{D}_{\text{init}}$
2: $\mathcal{C} \leftarrow \{\}$ {Initialize new collection}
3: **for** each $q \in \mathcal{Q}_{\text{amb}}$ **do**
4:     Obtain ambiguous entities $E$ and image $i$ for $q$
5:     Generate $K$ candidate sub-questions $\hat{Q} = \{\hat{q}_1, ..., \hat{q}_K\}$ by

$$\hat{q}_k = f_{\text{SG}}(c_{\text{SG}}, i; \theta_{\text{SG}}), \quad k = 1, \ldots, K$$

6:     **for** each $\hat{q}_k \in \hat{Q}$ **do**
7:         Compute confidence score by

$$\left\langle s_k, p_{\text{yes}}^k \right\rangle = f_{\text{SV}}(c_{\text{SV}}, i; \theta_{\text{SV}}).$$

8:         **if** $p_{\text{yes}}^k \geq \tau$ **then**
9:             Add $(q, \hat{q}_k)$ to $\mathcal{C}$
10:        **end if**
11:    **end for**
12: **end for**
13: $\mathcal{D} \leftarrow \mathcal{D}_{\text{id\_init}} \cup \mathcal{C}$
14: **return** final dataset $\mathcal{D}$

---

After that, sub-questions are generated for every ambiguous question using the sub-question generator and the validator. That is, for each $q \in Q_{amb}$, the sub-question generator first generates $\hat{Q} = \{\hat{q}_1, ..., \hat{q}_K\}$, a set of candidate sub-questions. Assume that an ambiguous entity $e$ in $q$ has a set of instances $E = \{\bar{e}_1, \ldots \bar{e}_{|E|}\}$ in the image $i$, where only one instance is the ground-truth instance of $\bar{e}^*$. Then, $\hat{q}_k \in \hat{Q}$ is obtained by

$$\hat{q}_k = f_{\text{SG}}(c_{\text{SG}}, i; \theta_{\text{SG}}), \tag{1}$$

where $c_{\text{SG}}$ is a text prompt that includes information on the bounding boxes of all instances in $E$ and the highlighted target instance[2], and $\theta_{\text{SG}}$ is the fine-tuned parameters of LLaVA for sub-question generation. Since $f_{\text{SG}}(\cdot)$ is implemented by LLaVA, $K$ sub-questions are generated by adopting the nucleus sampling following the work of Sultan et al. (2020).

Although all $\hat{q}_i$'s in $\hat{Q}$ are generated by a VLM, some of them may not be identifiable. Therefore, the sub-question validator $f_{\text{SV}}(\cdot)$ filters out the unidentifiable sub-questions. For every $\hat{q}_k \in \hat{Q}$, its identifiability $s_k$ and its confidence $p_{\text{yes}}^k$ are computed by

$$\left\langle s_k, p_{\text{yes}}^k \right\rangle = f_{\text{SV}}(c_{\text{SV}}, i; \theta_{\text{SV}}). \tag{2}$$

Here, $s_k \in \{yes, no\}$ indicates whether $\hat{q}_k$ is valid enough to identify the target instance. As in the sub-question generator, $c_{\text{SV}}$ is a prompt that includes the bounding boxes of all instances of $E$ and the highlighted target instance, and $\theta_{\text{SV}}$ is a set of LLaVA parameters for sub-question validation. This binary label $s$ is derived from the predicted probability of the *yes* token, denoted as $p_{\text{yes}}$, obtained from the output distribution of $f_{\text{SV}}(\cdot)$. To ensure the high quality of sub-questions, a confidence-based filtering is applied with a threshold $\tau$. That is, only the sub-questions with $p_{\text{yes}}^k \geq \tau$ are included in $C$, a set of automatically generated sub-questions. The number of sub-questions collected in this way is 109,336. When $\mathcal{D}_{id\_init}$ denotes a subset of $\mathcal{D}_{init}$ of which members are all identifiable sub-questions, the final GQA-Q2Q dataset is a union of $\mathcal{D}_{id\_init}$ and $C$. Thus, the total number of sub-questions in GQA-Q2Q is 135,846.

## 4.3 DATASET ANALYSIS

For human evaluation of GQA-Q2Q, 150 sub-questions are randomly sampled from both human-annotated and machine-annotated data, respectively. Table 2 shows the results of human evaluations. Each sub-question is rated by three human evaluators on a three-point scale[3] for disambiguity and fluency. The average scores for disambiguity are 2.66 for human-annotated sub-questions and 2.42

---

[2]Detailed explanation on prompts is given in Appendix A.2.

[3]This scale is further explained in Appendix A.3.

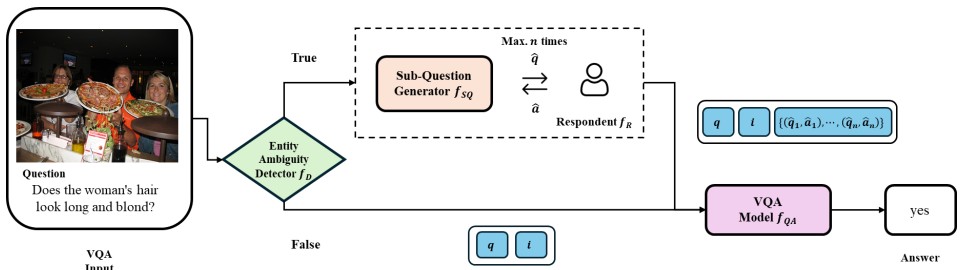

Figure 3: Overview of the proposed VQA framework for considering ambiguous questions.

for machine-annotated sub-questions. This result indicates that disambiguity is high for both types of sub-questions, and machine-annotated sub-questions are of similar quality to human-annotated ones. Fleiss' kappas and Krippendorff's alphas are over 0.4 and 0.6 for both types respectively, which implies moderate agreement among the evaluators. The average scores for fluency are also over 2.80 for both types. This result supports the linguistic plausibility of the sub-questions.

## 5 VISUAL QUESTION-ANSWERING WITH GQA-Q2Q

The validity of GQA-Q2Q dataset is shown by a VQA framework designed to leverage clarifying sub-questions to answer ambiguous questions. Figure 3 depicts the overall structure of the proposed VQA framework. Assume that a VQA sample $\langle q, i \rangle$ is given, where $q$ is a question and $i$ is an image related to $q$. The framework first applies the sample to the entity-ambiguity detector, $f_D(\cdot)$. For every entity $e$ appearing in $q$, $f_D$ determines the existence of its entity ambiguity by

$$y_{amb} = f_D(E),$$

where $y_{amb} \in \{true, false\}$ indicates the ambiguity existence of $e$ and $E$ is a set of instances of $e$ in the image $i$. In the paper, $f_D(\cdot)$ is a simple count-based classifier which labels $q$ as ambiguous if $|E| \geq 2$. That is, if any entity $e$ has two or more instances, then $q$ is ambiguous.

If $q$ is not ambiguous, a VQA model directly predicts an answer to $q$ using only $q$ and $i$, since $q$ does not contain any entity ambiguity. That is, the answer $a$ of $q$ is obtained by

$$a = f_{QA}(c_{qa}, i; \theta_{QA}), \tag{3}$$

where $f_{QA}(\cdot)$ is a VLM and $c_{qa}$ is a prompt about $q$. If $q$ is ambiguous, it is clarified by further asking sub-questions about the ambiguous entity in $q$. That is, for every entity instance $\bar{e}_k \in E$, the sub-question generator $f_{SG}(\cdot)$ re-trained with $\mathcal{D}$, GQA-Q2Q dataset, produces a clarifying sub-question $\hat{q}_k$ about $\bar{e}_k$ by Equation (1) for resolving the ambiguity of an instance $e$. Then, the respondent which is either an oracle or a model that is aware of the ground-truth target instance $\bar{e}^*$ provides a sub-answer $\hat{a}_k$ to $\hat{q}_k$. That is, when a respondent model, $f_R(\cdot)$, is a VLM, $\hat{a}$ is obtained by

$$\hat{a}_k = f_R(c_{resp}, i; \theta_{resp}), \tag{4}$$

where $c_{resp}$ is a prompt about $\hat{q}_k$, $\bar{e}^*$, and $E$, and $\theta_{resp}$ is the parameters for the VLM.

Note that $\hat{a}_k$ is *yes* or *no* since $\hat{q}_k$ is a yes/no question. It is *yes* if $\hat{q}_k$ asks about the features of the ground-truth instance $\bar{e}^*$, and is *no* otherwise. This process of obtaining $\hat{q}_k$ and $\hat{a}_k$ is repeated up to $n$ times to gather information about $\bar{e}^*$, where $n = |E|$. Thus, the collected pairs of a sub-question and its answer, $A = \{(\hat{q}_1, \hat{a}_1), \ldots, (\hat{q}_n, \hat{a}_n)\}$, are used as an input to enhance the reasoning of the ambiguous question $q$.

Finally, $f_{QA}(\cdot)$, a VQA model implemented with a VLM, generates a final answer $a$ from the ambiguous question $q$, the image $i$, the sub-question $\hat{q}$, and the set of sub-questions and sub-answers $A$ by

$$a = f_{QA}(c_{qa}^{amb}, i; \theta_{QA}), \tag{5}$$

where $c_{qa}^{amb}$ is a prompt explaining $q$ and $A$, and $\theta_{QA}$ is the VLM parameters for question-answering. When compared with Equation (3), $c_{qa}$ does not consider $A$ since $q$ is unambiguous in that case.

| Sub-question Generator | | | Sub-question Validator |
| --- | --- | --- | --- |
| BLEU | ROUGE-L | BERTScore | Accuracy |
| 45.03 | 61.15 | 91.29 | 82.81% |

Table 3: The performances of the sub-question generator and the validator on the initial GQA-Q2Q test set.

| VQA Model | Respondent | |
| --- | --- | --- |
| | Human Oracle | LLaVA |
| LLaVA-1.5$_{\text{Vicuna 7B}}$ | 66.03% | 65.61% |
| LLaVA-1.6$_{\text{Vicuna 7B}}$ | 70.15% | 69.49% |

Table 4: VQA accuracy comparison between a human respondent and LLaVA as a respondent.

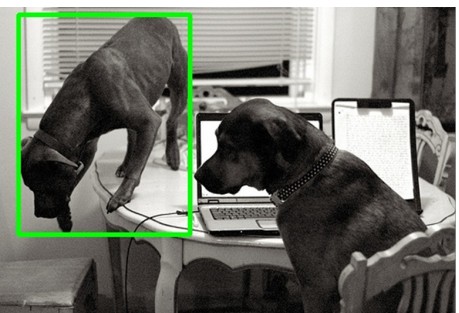

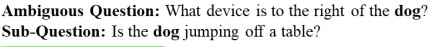

**Ambiguous Question:** What device is to the right of the **dog**?
**Sub-Question:** Is the **dog** jumping off a table?

Human Annotation

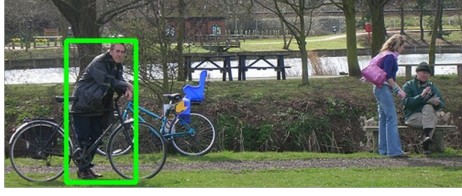

**Ambiguous Question:** What does the **man** lean against?
**Sub-Question:** Is the **man** wearing a black jacket?

Machine Annotation

Figure 4: Two samples of GQA-Q2Q dataset from human annotation and machine annotation.

# 6 EXPERIMENTS

## 6.1 EXPERIMENTAL SETTINGS

LLaVA-1.6$_{\text{Vicuna 13B}}$ is adopted as a backbone model for both the sub-question generator and sub-question validator in machine annotation. In addition, the LoRA (Hu et al., 2022) is adopted to finetune their parameters efficiently. In the VQA task to show the effectiveness of GQA-Q2Q dataset, LLaVA-1.6$_{\text{Vicuna 7B}}$ is used for $f_{\text{R}}(\cdot)$ in Equation (4) if a VLM is used as a respondent model, and BLIP2 (Li et al., 2023), InstructBLIP (Dai et al., 2023), LLaVa-1.5 (Liu et al., 2023), and LLaVA-1.6 (Liu et al., 2024) are tested as a candidate backbone for $f_{\text{QA}}(\cdot)$ in Equation (5). All experiments below are performed on a machine with eight RTX 6000 Ada generation GPUs.

The hyper-parameters $M$ and $K$ in Figure 2 are both set to five, while $\tau$ to control the quality of sub-questions in Algorithm 1 is set to 0.9[4]. The high value of $\tau$ ensures that only the sub-questions that receive strong agreement from the validator are included in GQA-Q2Q. In addition, the maximum number of interactions between $f_{\text{SG}}(\cdot)$ and $f_{\text{R}}(\cdot)$ is limited to the number of entity instances. That is, $n = |E|$ is used in Figure 3. Automatic evaluation metrics of BLEU, ROUGE, and BERTScore are adopted to assess the sub-question generator, while the performances of the sub-question validator and VQA models are evaluated with accuracy. More settings are explained in Appendix A.5.

## 6.2 EVALUATION OF GQA-Q2Q

The initial GQA-Q2Q dataset is used to train the sub-question generator and the validator used during machine annotation. Table 3 shows their performance after trained with the training set of the initial GQA-Q2Q dataset, where the performance is measured on the test set of the initial GQA-Q2Q. The question generator achieves 45.03 of BLEU, 61.15 of ROUGE-L, and 91.29 of BERTScore, which proves that the sub-question generator is trustworthy. On the other hand, the accuracy of the validator is only 82.81%. This is the reason why even the sub-questions that receive a '*yes*' answer from the validator should be filtered again using the hyper-parameter $\tau$. Further details on the effect of the hyper-parameter $K$ are provided in Appendix A.6.

Figure 4 shows two examples of the generated sub-questions. The left side of this figure comes from the initial set, where sub-questions are chosen by a human annotator. Here, the question mentions

---

[4]The reason why 0.9 is chosen is explained in Appendix A.4.

| Method | Accuracy (%) |
|---|---|
| BLIP-2$_{\text{Flan-T5 XL}}$ | 40.23 |
| + RepARe (Prasad et al., 2024) | 46.36 |
| + Proposed Framework | 42.56 |
| + (RepARe + Proposed Framework) | **46.53** |
| InstructBLIP$_{\text{Flan-T5 XL}}$ | 44.77 |
| + RepARe | 49.67 |
| + Proposed Framework | 48.69 |
| + (RepARe + Proposed Framework) | **50.32** |
| LLaVA-1.5$_{\text{Vicuna 7B}}$ | 61.03 |
| + Proposed Framework | **63.38** |
| LLaVA-1.6$_{\text{Vicuna 7B}}$ | 64.95 |
| + Proposed Framework | **67.79** |

Table 5: VQA performances of the answering models with the proposed sub-question generator.

an ambiguous entity '*dog*', but the sub-question distinguishes the target dog from the other dog by asking its attribute '*jumping off a table*'. Similarly, the right side is sampled from the machine-annotated sub-questions. This sub-question also identifies the target instance of the ambiguous entity '*man*' by mentioning the attribute '*black jacket*'. More examples are given in Appendix A.7.

### 6.3 EVALUATION OF VQA WITH AMBIGUOUS QUESTIONS

Table 4 compares the VQA accuracies with a human respondent and LLaVA as a respondent. To evaluate them automatically, the identifiable sub-questions of $\mathcal{D}_{init}$ are used instead of the sub-question generator in Figure 3, since the labels of the sub-questions in $\mathcal{D}_{init}$ are already known. According to this table, the accuracies of LLaVA are close to those of a human respondent, regardless of which LLaVA version is used as a backbone answering model. This implies that LLaVA is as reliable as a human oracle and thus the proposed VQA framework can be completely automated by replacing the human respondent with LLaVA.

Table 5 proves that the proposed VQA framework is effective to enhance various VQA backbone models. All accuracies in this table are achieved by using LLaVA as a respondent model, and they are measured on the test set of the initial GQA-Q2Q. The proposed framework outperforms across all backbone models with improvements of 2.33% for BLIP2, 3.92% for InstructBLIP, 2.35% for LLaVA-1.5, and 2.84% for LLaVA-1.6, which indicates that the proposed VQA framework is effective in improving VQA performance on ambiguous questions, since the proposed generated sub-question and sub-answer resolve the entity ambiguity of ambiguous questions. Recall that RepARe rewrites ambiguous questions to unambiguous ones by considering various visual information. It also improves the accuracy of BLIP2 and InstructBLIP. However, RepARe and the proposed framework are not orthogonal. That is, even after RepARe rewrites an ambiguous question, the proposed framework can still resolve the entity ambiguity of the rewritten question. In this way, the accuracy of RepARe is improved for both BLIP2 and InstructBLIP. Nevertheless, the best accuracy of 67.79% is achieved when LLaVA-1.6$_{\text{Vicuna 7B}}$ is used with the proposed framework. All these results demonstrate that the proposed framework is distinct and effective in resolving entity ambiguity.

### 7 CONCLUSION

This paper presented the first large-scale benchmark for resolving entity ambiguity in VQA through targeted sub-question generation. Based on the definition of ambiguous questions, GQA-Q2Q, a new dataset of clarifying sub-questions, was constructed from GQA benchmark. To achieve both scalability and high quality of the dataset, a human-machine collaborative pipeline was developed that combines template-based candidate generation, human verification, and model-driven data augmentation. In addition, a novel VQA framework was proposed to evaluate the validity of the proposed dataset. This framework explicitly incorporates ambiguity detection and resolution by utilizing the generated sub-questions and their corresponding answers before generating a final answer. The experimental results showed that leveraging the proposed GQA-Q2Q dataset leads to more accurate answers in scenarios involving entity ambiguity.

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

# A  APPENDIX

| Type | Template | Example |
|------|----------|---------|
| Attribute | Is/Are \<target entity\> \<attribute value\>? | *Is the shirt white?* |
| Relation | Is/Are \<target entity\> \<relation value\> \<object\>? | *Are the people located to the left of a pole?* |
| Position | Is/Are \<target entity\> on the left of the other \<entity\>? | *Is the shirt on the left of the other shirts?* |
| | Is/Are \<target entity\> on the right of the other \<entity\>? | *Is the wine glass to the right of the other wine glasses?* |
| | Is/Are \<target entity\> in the middle of the other \<entity\>? | *Is the man in the middle of the other men?* |

Table 6: Template patterns for generating candidate sub-questions in human annotation and their examples.

## A.1  TEMPLATES FOR GENERATING CANDIDATE SUB-QUESTIONS IN HUMAN ANNOTATION

The human annotation in Figure 2 is based on the candidate sub-questions generated automatically with templates. The purpose of these sub-questions is to distinguish the target instance of an ambiguous entity from other instances in the image. To achieve this, five templates for sub-question generation are devised, where each template incorporates a unique property of the target instance. Table 6 enumerates the templates and their examples. This paper considers three main types of properties for the templates: attributes of an ambiguous entity, relations between entities, and their spatial positions. The attribute-based sub-questions inquire about distinctive features of the ambiguous entity, such as its color, and the relation-based sub-questions describe either spatial or semantic relations between the ambiguous entity and other unambiguous entities. The final position-based sub-questions localize the target instance of the ambiguous entity in relation to the relative locations of other instances. In order to generate candidate sub-questions from the templates, the variables in the templates are filled automatically by comparing the target instance with other instances or the target entity with other entities in the scene graphs given in GQA dataset. Since the candidate sub-questions are generated based on the scene graphs, they are informative and effective, enabling a human annotator to efficiently select or refine high-quality sub-questions.

## A.2  PROMPTS FOR LLAVA

This paper adopts LLaVA as a backbone model for the sub-question generator, the sub-question validator in machine annotation, and the respondent and the final answering model in VQA. LLaVA is a VLM capable of generating a text response based on both a textual prompt and an input image. Each module devises its own text prompt under the consideration of the instruction-following patterns LLaVA was trained with in order to make the LLaVA perform appropriately in the targeted task.

Let $E = \{\bar{e}_1, \ldots, \bar{e}_{|E|}\}$ be a set of entities appearing in the input image. The sub-question generator aims to output identifiable candidate sub-questions that distinguish the target entity instance among ambiguous instances. To generate such candidate sub-questions, spatial locations of the entities represented by bounding boxes are essential for extracting the unique property of the target entity. Thus, the prompt for the sub-questions generator, $c_{\mathrm{SG}}$ in Equation (1), is designed to be

---

$< \mathrm{entity}_1 >: [x_1^{\bar{e}_1}, y_1^{\bar{e}_1}, x_2^{\bar{e}_1}, y_2^{\bar{e}_1}],$

$\quad \ldots,$

$< \mathrm{entity}_{|E|} >: [x_1^{\bar{e}_{|E|}}, y_1^{\bar{e}_{|E|}}, x_2^{\bar{e}_{|E|}}, y_2^{\bar{e}_{|E|}}]$

Target Entity: $< \mathrm{entity}_t >: [x_1^{\bar{e}_t}, y_1^{\bar{e}_t}, x_2^{\bar{e}_t}, y_2^{\bar{e}_t}]$

Generate a sub-question to classify ambiguous entities.

Sub-Question:

---

where $x$ and $y$ are the coordinates of an entity and $\bar{e}_t$ represented as $< \mathrm{entity}_t >$ is a target entity. During dataset construction, $\bar{e}_t$ is the ground-truth entity instance $\bar{e}^*$ during dataset construction, and is one of the entity instances in $E$ in the VQA framework.

Similarly, the goal of sub-question validator is to determine whether a generated candidate can identify the target entity effectively. Therefore, its prompt is designed to include spatial locations of all instances of an ambiguous entity, the target instance, and the sub-question. Consequently, when $I = \{\bar{e}_1, \ldots, \bar{e}_{|I|}\}$ is a set of instances of an ambiguous entity, the prompt for the sub-question validator, $c_{\mathrm{SV}}$ in Equation (2), is

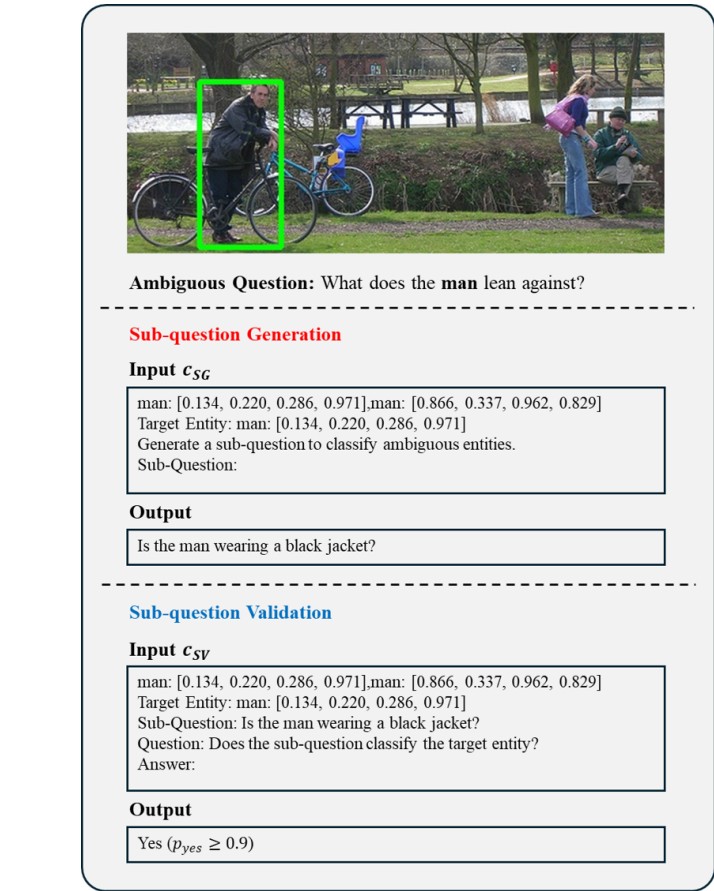

Figure 5: An example of the prompt $c_{SG}$ and $c_{SV}$ and their corresponding outputs during machine annotation.

$$< \text{entity}_1 >: [x_1^{\bar{e}_1}, y_1^{\bar{e}_1}, x_2^{\bar{e}_1}, y_2^{\bar{e}_1}],$$
$$\dots,$$
$$< \text{entity}_{|I|} >: [x_1^{\bar{e}_{|I|}}, y_1^{\bar{e}_{|I|}}, x_2^{\bar{e}_{|I|}}, y_2^{\bar{e}_{|I|}}]$$

Target Entity: $< \text{entity}_t >: [x_1^{\bar{e}_t}, y_1^{\bar{e}_t}, x_2^{\bar{e}_t}, y_2^{\bar{e}_t}]$
Sub-Question: $<$sub-question$>$
Question: Does the sub-question classify the target entity?
Answer:

where $<$sub-question$>$ is a sub-question generated by the sub-question generator.

Figure 5 shows an actual usage of $c_{SG}$ and $c_{SV}$. In this figure, the ambiguous entity is '*man*', since there are two men in the image. Thus, the information about the coordinates of the two men and the target instance highlighted with a bounding box is included in both $c_{SG}$ and $c_{SV}$. The difference between $c_{SG}$ and $c_{SV}$ is that a sub-question generated by the sub-question generator is provided at the end of the information in $c_{SV}$ while $c_{SG}$ has just an instruction. In this figure, the validator assigns the confidence score $p_{yes}$ of 0.999 to the sub-question "*Is the man wearing a black jacket?*" generated by the sub-question generator. Since $p_{yes} > 0.9$, the sub-question is accepted as an identifiable sub-question.

The respondent in Figure 3 is a model that is aware of the ground-truth target instance. It outputs a sub-answer, either '*yes*' or '*no*', where '*yes*' indicates the sub-question is correctly asking about the ground-truth target instance and the target instance has the property described in sub-question. Conversely, '*no*' implies that the sub-question does not refer to the target instance or the target instance does not have the described

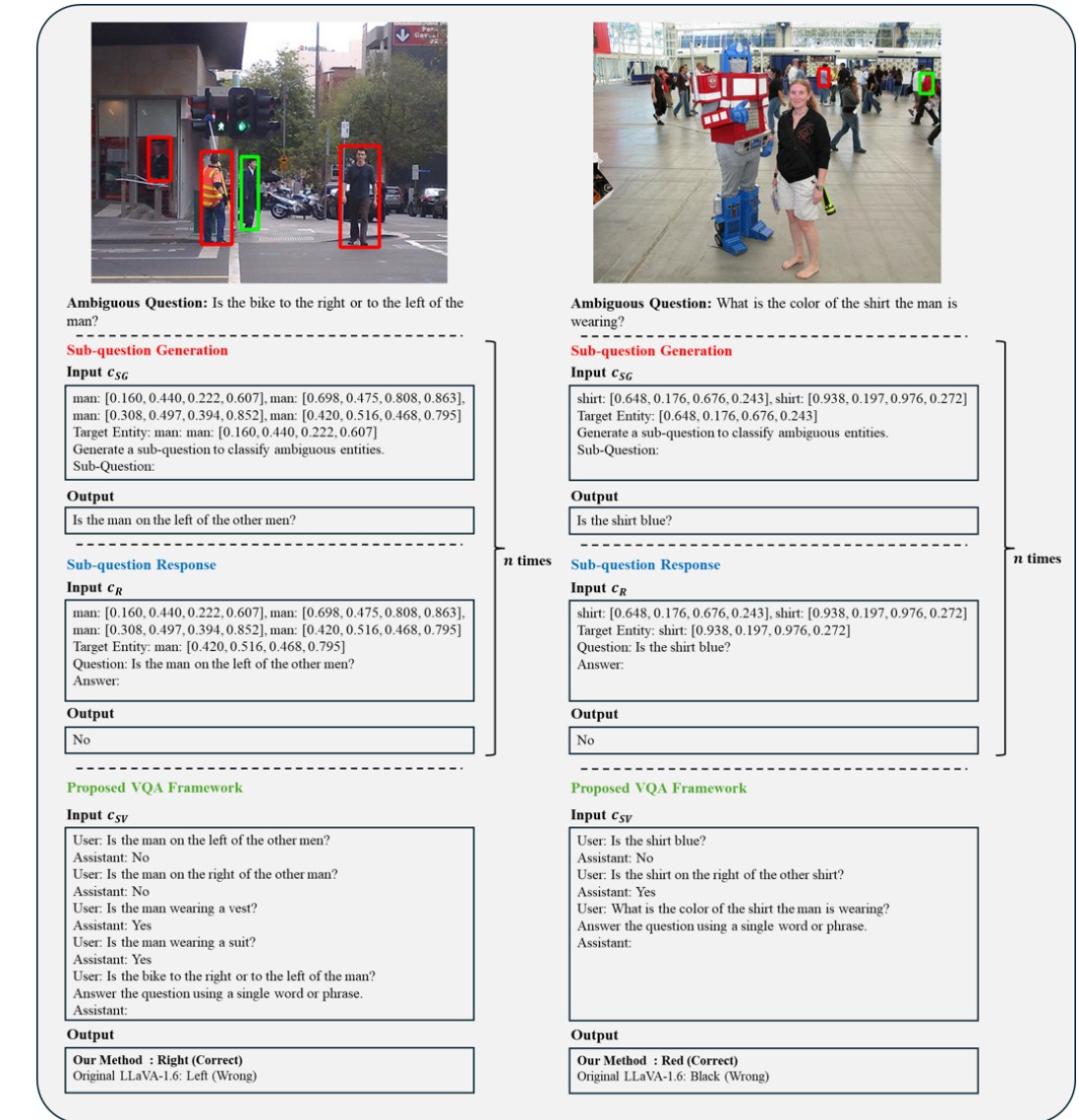

Figure 6: Two examples of the prompts $c_{\text{SG}}$, $c_{\text{R}}$, and $c_{\text{QA}}$ and their corresponding outputs during VQA. In the images, the green bounding box indicates the ground-truth entity instance, while the red bounding boxes denote other ambiguous entity instances, which are visualized only for illustration and not provided as input to the model.

property. Therefore, the prompt for the respondent, $c_{\text{R}}$ in Equation (3), is as follows.

$< \text{entity}_1 >: [x_1^{\bar{e}_1}, y_1^{\bar{e}_1}, x_2^{\bar{e}_1}, y_2^{\bar{e}_1}],$
    $\dots,$
$< \text{entity}_{|E|} >: [x_1^{\bar{e}_{|E|}}, y_1^{\bar{e}_{|E|}}, x_2^{\bar{e}_{|E|}}, y_2^{\bar{e}_{|E|}}]$
Target Entity: $< \text{entity}_t >: [x_1^{\bar{e}_t}, y_1^{\bar{e}_t}, x_2^{\bar{e}_t}, y_2^{\bar{e}_t}]$
Question: $<$sub-question$>$
Answer the question as Yes or No.

In Figure 5, the final answering model generates an answer based on an image, an ambiguous question, and $n$ pairs of sub-questions and sub-answers. In the experiments, LLaVA-1.6$_{\text{Vicuna 7B}}$ achieves the hightest accuracy among the backbone models such as the BLIP-2, InstructBLIP, and LLaVA-1.5. Following the instruction-tuning paradigm of LLaVA and chat templates from Vicuna,

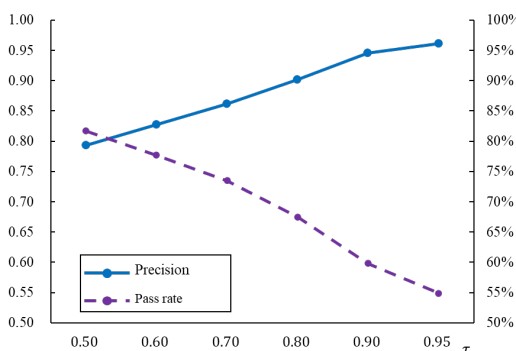

Figure 7: The change of the precision and the retention rate of identifiable sub-questions according to the values of $\tau$.

the prompt $c_{QA}$ is designed to guide LLaVA-1.6$_{\text{Vicuna 7B}}$ to reason step-by-step using the context of

> User: <sub-question 1>
> Assistant: <sub-answer 1>
> . . .
> User: <sub-question $n$ >
> Assistant: <sub-question $n$ >
> User: <question>
> Answer the question using a single word or phrase.
> Assistant:

where <question> is the ambiguous question and each pair of a sub-question and a sub-answer, repeated $n$ times, is generated by the sub-question generator and the respondent, respectively. By structuring the prompt in this multi-turn dialogue format, the LLaVA-1.6 is encouraged to perform a step-by-step reasoning before arriving at the final answer.

Figure 6 illustrates the entire process of the proposed VQA framework using the prompts $c_{\text{SG}}$, $c_{\text{R}}$, and $c_{\text{QA}}$. Three sub-questions are generated by the sub-question generator using $c_{\text{SG}}$, and they are all determined as *identifiable* by the respondent using $c_{\text{R}}$. Thus, $c_{\text{QA}}$ contains all these sub-questions and sub-answers as well as the original ambiguous question. LLaVA-1.6 using $c_{\text{QA}}$ gives a correct answer, whereas the original LLaVA produces an incorrect answer.

### A.3 HUMAN EVALUATION METRICS

To evaluate the sub-questions in the dataset qualitatively, a three-point scale test for disambiguity and fluency is adopted. Disambiguity assesses how effectively a sub-question identifies the target entity instance among ambiguous entity instances. A score of one implies that the sub-question does not refer to the correct target entity at all, A score of two denotes that it is effective for multiple instances including the target instance, and a score of three is assigned only when the sub-question clearly distinguishes the target instance from others.

### A.4 CONFIDENCE-BASED FILTERING

During machine annotation, a confidence-based filtering is used for the sub-question validator to select only the high-quality sub-questions from candidate sub-questions. The accuracy of the sub-question validator on the test split of the initial GQA-Q2Q dataset is 82.81% (see Table 3), but this is not high enough to exclude all unidentifiable sub-questions. Thus, a confidence threshold $\tau$ is applied to the result of the validator. That is, the sub-questions of which confidence is lower than $\tau$ are excluded from the final GQA-Q2Q dataset even if they are determined to be identifiable by the validator.

The precision and the pass rate are considered to find an optimal value of $\tau$, where the precision evaluates how accurately the validator identifies actual identifiable sub-questions and the pass rate indicates how much percentage of the ambiguous questions have at least one sub-question that passes this filter. With a high value for $\tau$, no candidate sub-question of some ambiguous questions can pass

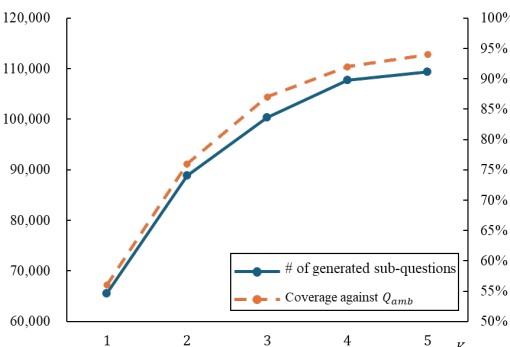

Figure 8: Change of the number of machine annotated sub-questions and their coverage against $\mathcal{Q}_{\text{amb}}$ according to $K$.

this filter, and then such ambiguous questions cannot be included in GQA-Q2Q. Figure 7 shows how the precision and the pass rate change according to $\tau$, based on measurements taken from the test set of the initial GQA-Q2Q. The precision has a trade-off relation with the pass rate. Thus, the precision climbs up but the pass rate drops down, as $\tau$ increases. In addition, note that the gradient of the precision gets smaller from $\tau = 0.9$. Therefore, to achieve both the scalability and the quality of the final dataset, $\tau = 0.9$ is used in all experiments.

### A.5 DETAILS OF EXPERIMENTAL SETTINGS

The sub-question generator and sub-question validator uses LLaVA-1.6$_{\text{Vicuna 13B}}$ as a backbone VLM. They are fine-tuned on initial GQA-Q2Q dataset with five epochs, a batch size of 128, and a learning rate of 2e-4. To enable efficient parameter finetuning, LoRA is applied with a rank of $r = 128$ and ab alpha of $a = 256$. Adam optimizer is adopted to train them without weight decay and with a cosine learning with a warm-up ratio of 0.03. All trainings are conducted on a machine equipped with eight Ada generation GPUs of type RTX 6000, and two RTX 6000 Ada generation GPUs are used at the inference time of the sub-question generator and the validator. All source codes for training and inference are customized following the official source codes of LLaVA[5] and RepARe[6]. The source codes and datasets will be publicly available after the reviewing process.

### A.6 EFFECT OF $K$ ON SUB-QUESTION GENERATION

Figure 8 depicts how the number of collected sub-questions is affected as the hyper-parameter $K$ increases. The bigger $K$ is, the more candidates the sub-question generator provides to the validator. Thus, as $K$ increases, the more sub-questions are collected. However, the difference between $K = 4$ and $K = 5$ is small, which implies that the number of collected sub-questions would not increase though $K$ is greater than five. Furthermore, when $K = 5$, sub-questions are generated from over 94% of the ambiguous questions in $\mathcal{Q}_{\text{amb}}$. This is the reason why $K = 5$ is used in the experiments.

Fluency evaluates the linguistic quality of a text. It assesses whether the text is grammatically correct, coherent, and natural. A score of one for fluency means that a sub-question is ungrammatical or awkward. A score of two indicates that a sub-question understandable but slightly unnatural, while a score of three implies that a sub-question is fluent and well-formed.

### A.7 ADDITIONAL EXAMPLES OF GQA-Q2Q

Following the samples shown in Figure 4, Figure 9 presents four additional samples of GQA-Q2Q, where two of them come from human annotation and the other two are sampled from machine annotation. In the human-annotated sample of the upper layer, the entity '*man*' in the ambiguous question is unclear, since there are a number of men in the image. However, only one man is hitting

---

[5]https://github.com/haotian-liu/LLaVA
[6]https://github.com/archiki/RepARe

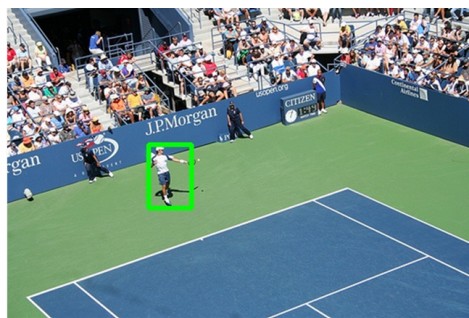

**Ambiguous Question:** What is the **man** wearing?
**Sub-Question:** Is the **man** hitting a ball?

Human Annotation

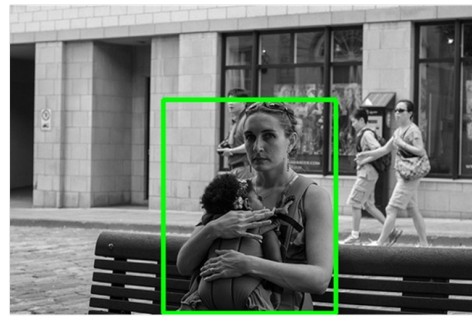

**Ambiguous Question:** Is the purse to the left or to the right the **woman**?
**Sub-Question:** Is the **woman** sitting on a bench?

Machine Annotation

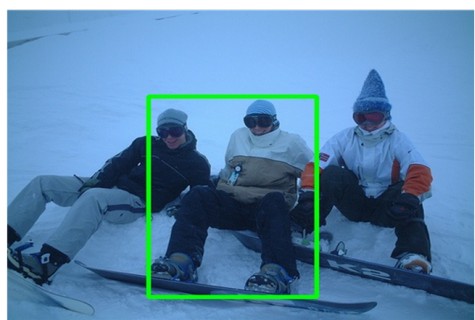

**Ambiguous Question:** Are the red glasses to the right of the **guy**?
**Sub-Question:** Is the **guy** in the middle of the other guys?

Human Annotation

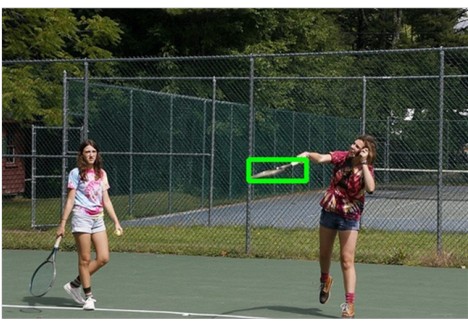

**Ambiguous Question:** Is the lady to the left or to the right of the **racket**?
**Sub-Question:** Is the **racket** to the right of the other racket?

Machine Annotation

Figure 9: Four additional samples of GQA-Q2Q dataset.

| Dataset | Split | No. of ambiguous questions | No. of questions |
|---------|-------|----------------------------|------------------|
| VizWiz | Validation | 353 | 4,319 |
| VQAv2 | Testdev | 21,009 | 107,394 |
| GQA | Train | 125,854 | 943,000 |

Table 7: Statistics of ambiguous questions across three VQA benchmarks.

a ball. Thus, the sub-question of "*Is the man hitting a ball*?" becomes an identifiable one. Similarly, in the machine-annotated sample of the upper layer, the entity '*woman*' in the ambiguous question becomes an ambiguous entity, since there are a woman sitting on a bench and two women walking down the street. Therefore, the sub-question "*Is the woman sitting on a bench*?" directly distinguishes the target instance of the '*woman*' entity from other instances. The sub-questions of the bottom layer are also effective in identifying the ambiguous entities of '*guy*' and '*racket*'.

## A.8 AMBIGUITIES OF VQA BENCHMARKS

Table 7 presents the number of ambiguous questions in three widely used VQA benchmarks. To identify the entities mentioned in each question for VizWiz (Gurari et al., 2018) and VQA-v2 (Goyal et al., 2017), this paper employs 'Florence-2-Large' from huggingface[7], which has demonstrated strong performance in object-centric grounding tasks (Xiao et al., 2024). The VizWiz validation set contains 353 ambiguous questions out of 4,319 (**8.17%**). The VQA-v2 testdev set contains 21,009 ambiguous questions out of 107,394 (**19.56%**). For GQA, 125,854 out of 943,000 training questions (**13.34%**) are ambiguous under our entity-level criterion.

---

[7]https://huggingface.co/microsoft/Florence-2-large

These results indicate that entity-level ambiguity is a general phenomenon across VQA benchmarks. Although VizWiz exhibits a lower proportion of ambiguous questions compared to GQA and VQA-v2, this is expected: VizWiz consists of images captured by blind people, where questions often refer to a specific object or detail, resulting in fewer multi-instance ambiguities. In contrast, VQA-v2 contains a higher proportion of entity-ambiguous questions, demonstrating that GQA is not an outlier in terms of ambiguity frequency. GQA provides an appropriate and representative benchmark with the necessary grounding annotations to a large-volume of entities, making it suitable for constructing and evaluating our ambiguity-resolution pipeline.

## A.9    Applicability to Other VQA Datasets

This paper develops and evaluates the ambiguity-resolution using GQA, a large-scale real-world VQA dataset whose scenegraphs and questions are generated through a fully automatic pipeline derived from Visual Genome annotations. This sub-question generation process demonstrates that entity-level grounding can produced algorithmically rather than manually, which is crucial for reproducibly identifying ambiguous instances.

The proposed pipeline is not restricted to GQA. Its applicability depends solely on the availability of entities referenced by the questions. Based on this requirement, the framework can be applied to other VQA datasets in two scenarios: (i) a dataset that contains grounding annotations, or (ii) a dataset without explicit grounding annotations. If grounding annotations are available, the pipeline can be applied directly. The required entities can be obtained automatically by generating scene graphs using grounding models trained on Visual Genome, as in GQA. If applying Visual Genome methods is challenging, a VLM model such as 'Florence-2-Large' (Xiao et al., 2024) can be used to extract entities from questions. Once a target entity is identified, the remainder of our framework operates unchanged. Overall, this paper proposes a dataset-agnostic pipeline that can be readily applied across a broad range of VQA benchmarks.

