# OpenReview forum: "GQA-Q2Q: A Large-scale Dataset for Resolving Entity Ambiguity in Visual Question-Answering via Clarifying Subquestion"
_ICLR.cc/2026/Conference — Submitted to ICLR 2026_

### Official Review · Reviewer_NciQ · 2025-10-30

**Soundness:** 3
**Presentation:** 3
**Contribution:** 3
**Rating:** 6
**Confidence:** 3

**Summary:**

The paper introduces GQA-Q2Q, a 135K-scale dataset of yes/no clarifying sub-questions designed to resolve entity-level referential ambiguity in the GQA VQA benchmark. It proposes a human-machine pipeline for scalable, high-quality data construction and validates utility via a multi-stage VQA framework that detects ambiguity, generates sub-questions, and conditions final answers on sub-answers. The work is timely, technically solid, and fills a clear gap in VQA ambiguity handling. While not revolutionary in method, the dataset scale, quality control, and downstream gains make it a strong candidate for acceptance.

**Strengths:**

1.	The paper is well-structured and easy to follow.
2.	The paper introduces GQA-Q2Q, the first large-scale dataset specifically designed to resolve entity-level referential ambiguity in visual question answering.
3.	The paper proposes a modular, interpretable VQA framework with explicit ambiguity resolution.

**Weaknesses:**

1.	Reliance on GQA Scene Graphs Limits Generalization: The sub-questions construction pipeline on more VQA dataset need to be further disscused.
2.	The authors contend that "the construction of a large-scale, high-quality dataset of clarifying sub-questions is essential for training and evaluating VQA models capable of handling ambiguous entities" (Lines 52–53), which is precisely the core motivation behind building this dataset. However, this claim would be strengthened by empirical experiments demonstrating that training on existing ambiguous datasets leads to suboptimal performance, and evaluation on such datasets may yield misleading results—for instance, models with clearly divergent capabilities exhibiting similar performance scores due to ambiguous entities.
3.	Is the proposed VQA framework designed to work specifically with the GQA dataset? Given that ambiguous entities exist across many existing datasets, if the framework's applicability is limited solely to GQA, it would significantly diminish the generalizability and impact of the proposed method.

**Questions:**

Please see the Weaknesses above.

---

> ### Author Response · Authors · 2025-11-20
>
> We appreciate the reviewer's concern. We hope our responses address the reviewer‘s concerns.
> ### NciQ W1 & W3
> The proposed framework does not require oracle-provided instance labels during inference; a standard VLM generates sub-answers without access to ground-truth entities, and the oracle annotations are used solely for the evaluation of VQA models. As detailed in the newly added Appendix A.9, our method is dataset-agnostic and relies only on identifying the entity referenced by the question. This can be done directly when grounding annotations exist. For datasets without such annotations, the entities can be obtained automatically by constructing scene graphs using grounding models trained on Visual Genome as in the GQA dataset or by employing modern grounding models such as Florence-2-Large. Once the entities are identified, the ambiguity-resolution process proceeds through our pipeline exactly as in the GQA setting. Thus, the approach does not rely on privileged information and naturally generalizes beyond GQA.
>
> ### NciQ W2
> We empirically demonstrated that ambiguity negatively affects VQA performance. As shown in Table 5, both LLaVA-1.5 and LLaVA-1.6 exhibit substantially lower accuracy (61.03% and 64.95%) when answering ambiguous questions without any clarifying sub-questions. However, when entity-level ambiguity is resolved by providing sub-questions and sub-answers (Table 4 and Table 5), their accuracy increases noticeably (66.03% and 70.15 in the Human Oracle setting). The clear performance gap between the models without disambiguation and with disambiguation provides direct evidence that ambiguity degrades VQA performance and that resolving ambiguity leads to more reliable evaluations of model capability.
>
> In the revised version, we further include a quantitative comparison with other VQA benchmarks in Appendix A.8. Using a unified entity-level ambiguity criterion, we analyze VizWiz, VQA-v2, and GQA. The proportions of ambiguous questions are 8.17% (VizWiz), 19.56% (VQA-v2), and 13.34% (GQA), showing that entity ambiguity is common across existing datasets.
>
> The reviewer’s suggestion to compare models trained on ambiguous datasets versus ambiguity-free datasets is insightful. However, current VLMs are instruction-tuned on large-scale corpora that already contain ambiguous questions, making it infeasible to obtain a model “trained without ambiguity” for a clean comparison. Despite this limitation, our experimental results clearly show that generating clarifying sub-questions effectively removes entity-level ambiguity and leads to measurable performance improvements. Thus, the validity and necessity of our ambiguity-resolution approach remain intact.

---

### Official Review · Reviewer_Vbg5 · 2025-10-30

**Soundness:** 3
**Presentation:** 3
**Contribution:** 3
**Rating:** 6
**Confidence:** 5

**Summary:**

This paper addresses the critical issue of entity ambiguity in Visual Question Answering (VQA), where a question refers to an entity with multiple instances in an image, causing models to fail.
To tackle this, the authors introduce GQA-Q2Q, the first large-scale dataset containing 135,846 clarifying sub-questions designed specifically to disambiguate these entities.
The dataset was constructed using an innovative human-machine collaborative pipeline, which starts with a small, human-verified seed set to train a sub-question generator and validator for scalable, automated data creation.
The paper demonstrates the dataset's value through a novel VQA framework that leverages these sub-questions, showing significant accuracy improvements across multiple VQA models and proving its effectiveness over existing methods that do not explicitly resolve entity ambiguity.

**Strengths:**

1.  The paper introduces GQA-Q2Q, the first large-scale benchmark dataset specifically designed to resolve entity ambiguity in VQA through the generation of clarifying sub-questions. With 135,846 sub questions, this dataset addresses a significant gap in existing resources and provides a valuable asset for training and evaluating models on this specific challenge.
2. The work proposes an efficient human-machine collaborative pipeline to construct the dataset. This hybrid approach effectively balances quality and scale by first creating a high-quality initial set with human oversight and then using it to train a sub-question generator and validator to automate the large-scale annotation process.
3. The paper provides robust empirical evidence that integrating the proposed framework improves the accuracy of multiple strong VQA backbone models (e.g., BLIP-2, InstructBLIP, LLaVA) on ambiguous questions. Furthermore, the experiments demonstrate that the proposed method is complementary to, and can enhance, existing ambiguity resolution techniques like RepARe, highlighting its distinct and valuable contribution to the field.

**Weaknesses:**

1. The criterion for ambiguity detection is oversimplified. By defining ambiguity solely based on the co-occurrence of multiple entity instances, the approach fails to encompass a broader spectrum of more complex semantic ambiguities, such as those arising from referential expressions, unclear relationships, or attribute specifiers. This definition is explicitly stated in the main text.
2. The method of synthetically augmenting ambiguity, specifically by removing adjectival modifiers, risks introducing a distributional shift. The resulting ambiguities may not be representative of or equivalent to the subtleties and patterns of naturally occurring linguistic ambiguity.
3. The modest accuracy of the sub-question validator (82.81%) is a notable limitation, necessitating a high confidence threshold (τ) to mitigate the inclusion of unidentifiable sub-questions. While the appendix presents a trade-off analysis between precision and pass rate, a more systematic report on the end-to-end impact of this filtering strategy on the final quality and potential biases of the large-scale dataset is warranted.
3. The proposed VQA framework relies on a strong oracle assumption for its 'respondent' module, which is presumed to know the ground-truth target instance. This limits the framework's applicability in real-world, open-ended scenarios where such privileged information is unavailable.

[Minor]
1. The justification for the selected confidence threshold (τ=0.9) is based on dataset construction metrics, but the paper lacks a sensitivity analysis of how this choice impacts downstream VQA performance. An ablation study showing VQA accuracy across different values of τ would provide a more complete picture of the hyperparameter's influence and ensure consistent reporting between dataset quality and task performance.
2. The generalizability of the approach remains unclear, as it has not been evaluated on external VQA benchmarks known to contain naturally occurring ambiguities.

**Questions:**

Please refer to the Weakness section.

---

> ### Author Response · Authors · 2025-11-20
>
> We appreciate the reviewer's concern. We hope our responses address the reviewer‘s concerns.
>
> ### Vbg5 W1
> While we agree that referential or relational ambiguities are important, prior studies have primarily focused on different types of semantic ambiguity in VQA. For example, Chen et al. first introduced the notion of focus ambiguity, and Pu et al. examined linguistic ambiguity through clarification questions.
>
> By contrast, entity-level ambiguity is the only ambiguity type that can be systematically and reproducibly identified using existing grounding annotations in large VQA benchmarks. As described in Section 3, we defined ambiguity based on whether the referenced entity corresponds to multiple instances within an image.
>
> Although earlier studies acknowledged the existence of ambiguity in VQA, there has been no prior large-scale dataset or pipeline explicitly designed to resolve entity-level ambiguity.  Our work fills this gap by establishing a precise and scalable formulation that can serve as the basis for extending to broader semantic ambiguity categories in future work.
>
> **References**
>
> - Chongyan Chen, Yu-Yun Tseng, Zhuoheng Li, Anush Venkatesh, and Danna Gurari. Accounting for focus ambiguity in visual questions. In *Proceedings of the IEEE/CVF International Conference on Computer Vision*, pp. 1228-1238, 2025.
>
> - Pu Jian, Donglei Yu, Wen Yang, Shuo Ren, and Jiajun Zhang. Teaching Vision-Language Models to Ask: Resolving Ambiguity in Visual Questions. In *Proceedings of the 63rd Annual Meeting of the Association for Computational Linguistics*, pp. 3619–3638, 2025.
>
>
> ### Vbg5 W2. & M2
> Our synthetic augmentation strategy removes adjectival modifiers to isolate entity-level ambiguity, as adjectives often serve to uniquely identify a specific instance and thus eliminate the very ambiguity we aim to study. Importantly, removing these modifiers does not alter the original intent of the question; the resulting sentences remain grammatical and natural, and continue to express the same referential structure. To minimize distributional shift, we selectively remove only the adjectives directly modifying the target entity using a constituency parser (ptb-3-revised_electra-large), leaving the rest of the question intact.
>
> Regarding generalizability, the revised version includes an analysis at Appendix A.8 that compares naturally occurring ambiguous questions in VQA-v2, VizWiz, and GQA. This analysis shows that entity-level ambiguity arises organically across all these benchmarks, confirming that our formulation is not an artifact of synthetic modification but reflects a phenomenon broadly present in real-world VQA datasets. Our augmentation merely ensures controlled coverage of such cases, complementing rather than replacing naturally occurring ambiguity.
>
>
> ### Vbg5 W3 & M1
> We agree that a more systematic analysis of the validator’s confidence threshold (τ) would further strengthen the paper. We appreciate this comment, and based on the reviewer’s suggestion, we have begun exploring additional τ values. However, modifying τ requires *regenerating the entire set of sub-questions*, followed by *retraining the sub-question generator* to ensure consistency with the new supervision signals. This process is computationally expensive and would take several weeks with our available resources, making it infeasible to include within the December 3 deadline. If the paper is accepted, we plan to incorporate a full sensitivity analysis in the camera-ready version.
>
>
> ### Vbg5 W4
> The proposed framework does not require oracle-provided instance labels during inference; a standard VLM generates sub-answers without access to ground-truth entities, and the oracle annotations are used solely for evaluating VQA models. As detailed in the newly added Appendix A.9, our method is dataset-agnostic and relies only on identifying the entity referenced by the question. This can be done directly when grounding annotations exist. For datasets without such annotations, the entities can be obtained automatically by constructing scene graphs using grounding models trained on Visual Genome as in the GQA dataset or by employing modern grounding models such as Florence-2-Large. Once the entities are identified, the ambiguity-resolution process proceeds through our pipeline exactly as in the GQA setting.Thus, the approach does not rely on privileged information and naturally generalizes beyond GQA.

---

### Official Review · Reviewer_ru4P · 2025-10-31

**Soundness:** 1
**Presentation:** 2
**Contribution:** 2
**Rating:** 2
**Confidence:** 4

**Summary:**

This paper targets the problem of ambiguous questions in Visual Question Answering (VQA), where the visual target entity referenced by the question is not clearly specified. To address this, the authors introduce GQA-Q2Q, a new large-scale benchmark dataset designed to pair ambiguous questions with clarifying sub-questions. The dataset is constructed using a hybrid human–machine pipeline: a small human-annotated seed set is expanded automatically via a trained sub-question generator and validator.

**Strengths:**

1. Ambiguity in VQA is an underexplored yet critical challenge for real-world deployment of vision-language systems. Introducing a dedicated dataset for this issue is valuable for future research.
2. The experiments with many LLMs contribute to the understanding the current LLMs.

**Weaknesses:**

1. The paper does not clearly define what constitutes an “ambiguous” question. While examples are given, the formal rules or annotation criteria (e.g., whether ambiguity arises from multiple visual entities, vague attributes, or linguistic cues) are not detailed. A lack of precise definition undermines dataset impact.

2. The paper does not quantitatively or qualitatively compare GQA-Q2Q with existing datasets (e.g., GQA, VQAv2, or VizWiz). It would be informative to show the proportion of ambiguous questions already present in existing benchmarks and how GQA-Q2Q extends or complements them.

3. The experiments on current VLMs are not sufficiently rigorous to separate hallucination errors from ambiguity-related failures. Since large models often hallucinate facts even for clear questions, additional analysis is needed to differentiate whether observed errors stem from visual ambiguity or model bias.

4. While the dataset scale is impressive, there is little discussion of sub-question quality, such as linguistic fluency, correctness, or diversity after automatic expansion.

**Questions:**

NA

---

> ### Author Response · Authors · 2025-11-20
>
> We appreciate the reviewer's concern. We hope our responses address the reviewer‘s concerns.
>
> ### ru4P W1
> We clearly and explicitly defined in the paper what constitutes an ambiguous question, and our dataset construction procedure as well as our VQA pipeline are developed strictly according to this definition. In Introduction, we defined an ambiguous question as the one that lacks a clearly identifiable target entity required for accurate visual reasoning in VQA (L44–L46). In Section 3, we further formalized this by stating that a question is considered entity-ambiguous when its referenced entity corresponds to multiple instances within an image (L177–L180). This establishes a precise and operational criterion that guides all stages of our dataset creation. This definition clearly distinguishes our notion of ambiguity, entity-level ambiguity, from other types such as focus or linguistic ambiguity (L180–L182).
>
> ### ru4p W2
> In the revised version, we include a quantitative comparison with other VQA benchmarks at Appendix A.8. Using a unified entity-level ambiguity criterion, we analyze VizWiz, VQA-v2, and GQA. The proportions of ambiguous questions are 8.17% (VizWiz), 19.56% (VQA-v2), and 13.34% (GQA) (see Table 7), showing that entity ambiguity is common across existing datasets.
> This analysis also clarifies how GQA-Q2Q complements prior benchmarks: VizWiz contains fewer ambiguous questions due to its single-object–focused images, whereas VQA-v2 and GQA involve more complex scenes with multiple instances. GQA further provides grounding annotations necessary for systematically identifying ambiguous entities, making it a suitable base for constructing our ambiguity-resolution dataset.
>
> ### ru4P W3
> Our goal is to generate sub-questions that eliminate ambiguity, and the VQA task in our work is used primarily to demonstrate the **effectiveness** of our ambiguity-resolution framework rather than to maximize raw VQA accuracy. Nevertheless, the reviewer‘s concern regarding how we distinguish hallucination errors from ambiguity-induced failures is addressed, because it is a critical issue.
>
> We acknowledge that current VLMs often exhibit hallucination errors in VQA tasks. To isolate the effect of visual ambiguity from general model bias, we evaluated model performance specifically on ambiguous questions, and we controlled for hallucination by comparing the results with and without sub-question and sub-answer inputs (Table 5). This controlled setup enables us to attribute performance differences directly to ambiguity resolution rather than to incidental hallucination behavior.
>
> Furthermore, Table 4 shows that even when provided with human-oracle sub-questions and sub-answers, both LLaVA-1.5 and LLaVA-1.6 exhibit only limited improvement. This indicates that hallucination persists regardless of whether ambiguity is resolved, confirming that hallucination is an independent phenomenon rather than a primary source of failures.
>
> ### ru4P W4
> As described in Section 4.3, we conducted a thorough human evaluation to assess the linguistic fluency and disambiguation quality of both human- and machine-generated sub-questions.
> The results in Table 2 show that machine-generated sub-questions achieve high fluency (>2.80) and comparable disambiguation quality (2.42 vs. 2.66 for human), confirming that automatic expansion also preserves quality.

---

### Official Review · Reviewer_NAiF · 2025-11-02

**Soundness:** 2
**Presentation:** 2
**Contribution:** 2
**Rating:** 4
**Confidence:** 3

**Summary:**

This paper presents a large-scale VQA dataset that disambiguates unclear entities through clarifying sub-questions. It also proposes a VQA framework that uses these sub-questions to resolve ambiguity before generating a final answer.

**Strengths:**

- A large-scale VQA dataset that disambiguates unclear entities is presented.

**Weaknesses:**

- Ambiguity issues may not always be resolvable. For example, in Figure 1, generating a sub-question to clarify the ambiguous entity is impossible without the ground-truth answer—the question itself cannot distinguish which "happy man" it refers to. If the ground-truth answer is required to remove ambiguity, this approach may has limited utility at inference time.
- Why is fine-tuning a VLM necessary for the sub-question generator and validator? A powerful VLM could perform both tasks with appropriate prompts. Moreover, fine-tuning may cause overfitting when annotated data are limited.
- The experiments lack strong evidence of effectiveness. Table 5 suggests that most performance gains come from RpeARe (Prasad et al., 2024) rather than the proposed method.
- Some terms are not defined, such as GQA-Q2Q. What does it stand for?

**Questions:**

- If the ground-truth answer is required to remove ambiguity, how will this approach work well at inference time?
- Why is fine-tuning a VLM necessary for the sub-question generator and validator?
- How do the results effectively demonstrate that the proposed method works?

---

> ### Author Response · Authors · 2025-11-20
>
> We appreciate the reviewer’s concern. We hope our responses address the reviewer’s concerns.
>
> ### NAiF W1 & Q1
> Ground-truth answers are required when constructing GQA-Q2Q to ensure dataset quality, but they are not mandatory during VQA inference. In interactive VQA scenarios in which a questioner and an answer model communicate, the ground-truth answers are provided by the questioner – an ideal use case for the proposed model. If a human questioner is unavailable, a VLM can serve as a substitute.
>
> Actually, in our approach (Section 5), no ground-truth answers are required during inference, as the volume of questions makes employing a human respondent impractical. That is, a VLM such as LLaVA acts as the respondent and automatically generates sub-answers. Consequently, the ambiguity-resolution process operates fully automatically without any ground-truth supervision during inference.
>
> Table 4 shows that the performance gap between a human oracle and LLaVA as a respondent is small (e.g., LLaVA-16: 70.15% vs 69.49%). This demonstrates that VLM-based respondents can approximate oracle-level performance, making ground-truth supervision unnecessary at inference time. In this paper, the human oracle is used only to estimate an upper bound of VLM capability. Therefore, our framework enables completely automatic ambiguity resolution without requiring any ground-truth answer.
>
> Note that our objective is not to improve VQA model performance, but to generate clear and explicit sub-questions that eliminate entity-level ambiguity of an original question. Accordingly, the VQA task in our work is not intended as the primary focus; rather, it serves as a means to evaluate the effectiveness of the proposed dataset and framework. Nevertheless, we carefully designed our pipeline to reflect practical real-world constraints, ensuring that the method remains deployable in realistic scenarios.
>
> ### NAiF W2 & Q2
> Our goal is to generate yes/no sub-questions that resolve entity ambiguity. Although both in-context learning and fine-tuning are possible approaches, prior work indicates that fine-tuning is more reliable and less sensitive to prompt variability in tasks requiring  structured output and visual grounding [1]. Thus, given our task-specific dataset, fine-tuning is the more appropriate choice.
> We fine-tune both the generator and validator using LoRA, enabling efficient adaptation with a small annotated seed set. Although our dataset is smaller than the full GQA corpus, it is sufficient for task-specific tuning. We monitor validation performance and apply early stopping to prevent overfitting.
>
> Table 2 shows that the human evaluation scores of machine-annotated sub-questions (Disambiguity = 2.42, Fluency = 2.63) remain comparable to human-written ones (2.66 and 2.76, respectively). Furthermore, Table 3 confirms that the fine-tuned generator achieves strong automatic metrics (BLEU = 45.03, ROUGE-L = 61.15, BERTScore = 91.29), and the validator reaches 82.81% accuracy, indicating that the models generalize well without memorization.
>
> **References**
>
> [1] Haokun Liu, Derek Tam, Mohammed Muqeeth, Jay Mohta, Tenghao Huang, Mohit Bansal, and Colin Raffel. Few-Shot Parameter-Efficient Fine-Tuning is Better and Cheaper than In-Context Learning. In *Proceedings of the 36th International Conference on Neural Information Processing Systems*. 2022.
>
> ### NAiF W3 & Q3
> Table 5 demonstrates that our framework alone consistently improves accuracy across all VQA backbones, even for RepARe. Although BLIP-2 shows a comparatively smaller improvement, we attribute this to its relatively weaker reasoning capability, particularly in utilizing sub-questions and sub-answers for disambiguation. When combined with RepARe, further gains are achieved, indicating complementarity rather than dependency. RepARe focuses on linguistic reformulation, whereas our method addresses entity-level ambiguity through clarifying sub-questions, which is an orthogonal contribution. In summary, our framework reliably enhances the base backbone model, regardless of which model is adopted as a backbone.
>
> ### NAiF W4
> We clarified in Introduction of the revised version that GQA-Q2Q stands for “GQA – ambiguous Question to clarified Question”, explicitly describing the mapping from an ambiguous question to its corresponding clarifying sub-question.

---

### Meta-Review · Area_Chair_fP5y · 2026-01-10

**Summary:**

The paper tackles the problem of generating sub-questions to resolve “entity ambiguity” in VQA. It receives ratings of 4, 2, 6, 6: Reviewer NAiF (4), Reviewer ru4P (2), Reviewer Vbg5 (6), Reviewer NciQ (6). Major concerns:

1. Lack of motivation (Reviewer ru4P): Reviewer ru4P wants to see an analysis on entity-ambiguity rates (as defined in this paper) on existing VQA datasets.

2. Narrow/oversimplified definition of ambiguity (Reviewer Vbg5). A question is ambiguous if it refers to an object that appears multiple times in the image (e.g., asking about a man when there are multiple men in the image). Reviewer Vbg5 feels this definition of ambiguity is over simplified. Further, Reviewer Vbg5 feels that the actual method (removing adjectival modifiers) used to synthetically inject more ambiguity in the paper creates an even more narrow type of ambiguity.

3. Experimental rigor (Reviewer NAiF, Reviewer ru4P, Reviewer NciQ)

3.1 Reviewer NAiF asks “How do the results effectively demonstrate that the proposed method works?”.

3.2 Reviewer ru4P wants to see more results on sub-question quality and feels that experiments are not rigorous enough to separate model hallucination errors from actual ambiguity-related failures.

3.3 Reviewer NciQ asks if training on existing ambiguous datasets lead to suboptimal performance.

4. Reliance on ground-truth to sub-questions (or scene graphs) limits applicability/generalization (Reviewer NAiF, Reviewer Vbg5, Reviewer NciQ). Reviewers feel that access to oracle is required to resolve ambiguity during inference, making the problem intractable. Reviewers also want to see the proposed method being applied to other datasets than GQA.

5. Other concerns

Most performance gains come from RpeARe (Prasad et al., 2024) in Table 5 (Reviewer NAiF)

Modest accuracy of the sub-question validator (Reviewer Vbg5)

**Reviewer Concerns:**

[Not resolved] 1. The authors respond by including the results in Table 7, with 8.17% (VizWiz), 19.56% (VQA-v2), and 13.34% (GQA). However, the method used to obtain these numbers is unclear (whether this is human or machine verification, how they use Florence-2-Large to extract entities and how do they make sure these are correct.).

[Not resolved] 2. The authors’ response fails to convince the significance of this type of ambiguity.

[Somewhat resolved, problem with clarity] 3. The authors point to Table 2 with evaluation on fluency and disambiguation quality. They also argue maximizing raw accuracy is not the main goal but generating high-quality subquestions. They explain the results in Table 5 and Table 4 to all the reviewers. They believe these help distinguish between hallucination vs. ambiguity-related failures.

[Not resolved] 4. The authors argue they can use VLMs for this purpose but I do not understand here how VLMs would know the user’s intent.

**Reviewer Scores:**

Reviewer NAiF (4): To keep

Reviewer ru4P (2): To keep

Reviewer Vbg5 (6): To keep

Reviewer NciQ (6): To keep

---

### Decision · Program_Chairs · 2026-01-26

Reject